# Prevalence and Credibility of Nutrition and Health Claims: Policy Implications from a Case Study of Mongolian Food Labels

**DOI:** 10.3390/ijerph17207456

**Published:** 2020-10-13

**Authors:** Nyamragchaa Chimedtseren, Bridget Kelly, Anne-Therese McMahon, Heather Yeatman

**Affiliations:** 1School of Health and Society, University of Wollongong, Northfield Avenue, Wollongong, NSW 2522, Australia; bkelly@uow.edu.au (B.K.); amcmahon@uow.edu.au (A.-T.M.); hyeatman@uow.edu.au (H.Y.); 2Nutrition Department, National Centre for Public Health, Peace Avenue 17, Ulaanbaatar 13374, Mongolia

**Keywords:** claims, food, beverage, label, nutrition, health

## Abstract

Nutrition and health claims should be truthful and not misleading. We aimed to determine the use of nutrition and health claims in packaged foods sold in Mongolia and examine their credibility. A cross-sectional study examined the label information of 1723 products sold in marketplaces in Ulaanbaatar, Mongolia. The claim data were analysed descriptively. In the absence of national regulations, the credibility of the nutrition claims was examined by using the Codex Alimentarius guidelines, while the credibility of the health claims was assessed by using the European Union (EU) Regulations (EC) No 1924/2006. Nutritional quality of products bearing claims was determined by nutrient profiling. Approximately 10% (*n* = 175) of products carried at least one health claim and 9% (*n* = 149) carried nutrition claims. The credibility of nutrition and health claims was very low. One-third of nutrition claims (33.7%, *n* = 97) were deemed credible, by having complete and accurate information on the content of the claimed nutrient/s. Only a few claims would be permitted in the EU countries by complying with the EU regulations. Approximately half of the products with nutrition claims and 40% of products with health claims were classified as less healthy products. The majority of nutrition and health claims on food products sold in Mongolia were judged as non-credible, and many of these claims were on unhealthy products. Rigorous and clear regulations are needed to prevent negative impacts of claims on food choices and consumption, and nutrition transition in Mongolia.

## 1. Introduction

Lifestyle-related non-communicable diseases (NCDs) are the leading cause of global deaths, responsible for 71% of the 57 million global deaths in 2016. Almost eight in every ten deaths from NCDs occur in low- and middle-income countries (LMIC) [1]. Nutrition transition can result in higher rates of obesity and NCDs and is associated with shifts in diet, physical activity and other lifestyle changes that follow economic, demographic and epidemiological changes [2]. Changes in diet are one of the key characteristics of nutrition transition. Dietary changes include increased consumption of processed foods and shifts from traditional diets to Western pattern diets high in energy, sugars and fat [2]. Nutrition transition is a global phenomenon but is occurring much faster in LMICs [3]. LMICs are facing challenges in responding to nutrition transition and a faster growing burden of NCDs. These challenges relate to limited resources and time to adjust food policies to support healthy diets. Serious attempts to address the problem are limited to only a few countries [4].

Provision of accurate and sufficient information on the nutritional quality of food products is a key policy action for governments to support healthy diets, as recommended by the Codex Alimentarius Commission [5]. Claims are one form of nutrition labelling. Nutrition claims state, suggest or imply that a food has particular nutritional properties including but not limited to the energy value and to the content of protein, fat and carbohydrates, as well as the content of vitamins and minerals. Health claims refer to relationships between a food or a constituent of that food and health [6]. Nutrition labelling provides information to consumers about the nutritional content of foods and assists them in making healthier choices. It may also encourage product reformulation as food manufacturers seek to avoid making undesirable disclosures [7].

Claims on food labels should be truthful and not misleading [6]. However, food producers use claims for marketing purposes [8]. Claims can be misleading where they are present on foods deemed less healthy or when health claims are not scientifically substantiated [8]. Claims also can induce a “health halo” effect, by which they affect consumers’ perceptions of the overall healthfulness of foods. People are more likely to purchase products bearing claims and are not as restrained in their consumption [9].

Mongolia is an LMIC where little research on food labelling has been undertaken. Prior to shifting to a market economy in the early 1990s, Mongolia was under a centralised economy and had low levels of imported food products [10]. Consequently, Mongolian consumers are relatively unfamiliar with food labelling specifically and processed packaged food more generally. The country is experiencing rapid nutrition transition with commensurate NCD burdens. NCDs surpassed other causes of mortality in recent decades to become the leading cause of population mortality. Cardiovascular disease and cancer accounted for 60% of population deaths in 2017, compared to 58% in 1995 [11]. Of 15–49 years olds, 46.2% of women and 48.8% of men were overweight and obese in 2016, which represents an increase of 40% for women and 77% among men from 2010 levels [12].

In Mongolia, a new food labelling standard, MNS 6648:2016, which was largely based on the relevant Codex standards for food labelling [5,6,13], came to enforcement in 2018. Prior to this, there was effectively no regulation relating to nutrition and health claims on food packages. The previous guideline on nutrition labelling of 2007, which was an apparent translation of the Codex guidelines on nutrition labelling [5], lacked capability to provide proper regulation due to its poor translation (introducing errors) and voluntary nature. The new regulation of 2018 was progressive to the previous guideline as it stipulates mandatory nutrition labelling for all pre-packaged food products on the back or side of food packaging. Official label languages are Mongolian, Russian and English. Regulations relating to nutrition and health claims are still minimal in the new standard and include two main requirements: (1) the mandatory declaration of a nutrient when a nutrition or health claim is made, and; (2) the need for approval of health claims by a government-authorised organization. A definition of a nutrition claim was provided in the food labelling standard MNS 6648:2016, together with the requirement to declare the amount of the claimed nutrient. The standard also introduced the concept of scientific substantiation of health claims. However, the standard does not specify the types of nutrition and health claims that are permitted and lacks requirements regarding criteria for making claims [14].

Food labelling policy implementation, including for nutrition and health claims, has not been well studied in developing countries. Most evidence on the use of claims and their effects on diets are from developed countries [15,16,17]. The study aimed to determine the use of nutrition and health claims on packaged foods sold in Mongolia and examine the credibility of these claims. As food labelling regulations are currently in transition in this country, this study provides a critical baseline evaluation of the food labelling landscape to guide identification of areas of concern and provide a basis for assessing progress on policy implementation. Findings will be useful to other developing countries experiencing similar trajectories in the availability and population consumption of processed packaged foods in the absence of corresponding food labelling policies to guide healthier choices.

## 2. Materials and Methods

### 2.1. Data Collection and Coding

A survey of packaged food product labels was conducted in Ulaanbaatar, the capital city of Mongolia, during November and December 2017. University students studying nutrition, public health and nursing were engaged in data collection after undertaking training in the data collection tool. The students collected the label information of food products from supermarkets and grocery stores located throughout the city. They were instructed to collect the product information from any supermarket or grocery store at their convenience.

Approximately 100 student data collectors sampled food products from 50 food categories belonging to 11 major groups (Table A1). These food categories and subcategories were based on the food categories’ classification used in the household socio-economic survey of the National Statistics Office of Mongolia [18], which represented the common types of food products used by Mongolian households with some modifications to include other common types of processed food products. The pre-defined food categories were pre-tested in one supermarket by crosschecking them against the products placed on the shelves in the supermarket and missing food categories were added.

The food categories were assigned to the data collectors in order to avoid duplications and each student was asked to collect label photographs of at least 20 food products across all label language groups, capturing as many different brands as possible. They took photographs of product packaging and recorded details of label information, including the product’s name, category, brand, manufacturing country, label language and availability of nutrient declarations and claims. Students transferred electronic copies of the photographs to the lead author (NCh).

Photographs were coded by one person (NCh) for product name, type, manufacturing country, label language and the verbatim content of claims. If label photographs were of poor quality or did not fully capture the label, students were asked to retake photographs of the products and send them through, or the Internet was searched for images of the products.

### 2.2. Data Analysis

Data were entered into Microsoft Excel (2016) and converted into IBM SPSS Statistics for Windows, Version 23.0 (IBM Corp., Armonk, NY, USA) for analysis. The proportions of food products carrying nutrition and health claims and the rate of claims per 100 products (a number of claims per 100 products) were estimated for each food category. The rates of claims were compared by claim type and label language.

By the credibility of claims, we perceived trustworthiness and reliability of claims in terms of providing reliable and scientific evidence-based information to consumers, as well as providing supporting information on the content of claimed nutrients to back up the claimed nutritional characteristics or health effects of a product. The Codex guidelines and the claims regulation of the EU were used in the credibility analysis of claims as the current national food labelling standard (2018) did not contain criteria for making nutrition and health claims. Credibility of nutrition claims was determined by their compliance with the criteria of nutrient content claims established in the Codex guidelines for Use of Nutrition and Health Claims (CAC/GL 23-1997) [6]. Nutrition claims were considered credible if the value for the claimed nutrient was present and in correct amounts on the nutrient declaration. Health claims were assessed for their consistency with the list of acceptable claims of the EU Regulations (EC) No 1924/2006 [19]. The EU regulation was used because of the considerable share in the Mongolian food imports from EU countries [20]. Health claims were considered credible if they appeared in this list and were compliant with the criteria of nutrient content established for corresponding claims (Figure 1).

Products with nutrition and health claims were assessed for their healthiness by comparing their nutrient content against the WHO nutrient profile model for the Western Pacific Region (WPR) [21]. The purpose of the model is to restrict marketing of foods and non-alcoholic beverages to children and it is intended to differentiate between food and non-alcoholic beverages that are more likely to be part of a healthy diet from those that are less likely. The model consists a total of 18 food categories and marketing to children is prohibited for three categories (category 1—chocolate and sugar confectionary, energy bars and sweet toppings and desserts; category 2—cakes, sweet biscuits and pastries and sweet bakery products; and category 4c—energy drinks, tea and coffee). The nutrient content of the products was crosschecked against the nutrient thresholds for saturated fats, trans fatty acids, added sugar and sodium of the model. Products that exceeded any of the relevant thresholds were considered unhealthy.

The research was reviewed and approved by the Human Research Ethics Committee of University of Wollongong on 24 October 2017 (Project identification code: 2017/394).

#### Classification of Claims

Claim types were determined according to the Codex classifications [6]. In addition, therapeutic claims were included as a type of health claim (Table 1).

## 3. Results

### 3.1. Characteristics of Food Products Surveyed

Label photos of 1723 food products were collected and analysed. The sample included nearly equal numbers of products labelled in Mongolian and other languages. The products belonged to 17 of 18 food categories of the WHO nutrient profile model for the WPR (Table 2). One-third of the products contained nutrient profiles in the categories (1, 2 and 4c), for which marketing to children is prohibited.

### 3.2. Prevalence of Nutrition and Health Claims on Products

Overall, 9% (*n* = 149) of products carried at least one nutrition claim and 10% (*n* = 175) of products carried at least one health claim. The most prevalent claims were nutrition claims, nutrient function claims and therapeutic claims. The median numbers of nutrition and health claims were 2 claims per product, respectively (Table 3). It was common for the same product to carry more than one claim so that 50.3% of products with nutrition claims and 81% of products with health claims had more than one claim per product.

#### 3.2.1. Prevalence of Nutrition and Health Claims by Label Language

Products labelled in Mongolian had higher rates of claims than those labelled in other languages. The prevalence of claims was between 2.2 and 21.7 times higher for products labelled in Mongolian (*n* = 856) compared to other languages (*n* = 867). Per 100 products, the different rates of claims for Mongolian labels compared with labels in other languages were: reduction of disease risk claims 4.3 (*n* = 37) versus 0.2 (*n* = 2), other function claims 15.7 (*n* = 134) versus 1.6 (*n* = 14), therapeutic claims 16.8 (*n* = 144) versus 1.8 (*n* = 16) and nutrition claims 23.0 (*n* = 197) versus 10.5 (*n* = 91), respectively.

#### 3.2.2. Products Carrying Nutrition and Health Claims

Product categories with the highest percentages of products with at least one nutrition claim and with the highest rates of nutrition claims were dried curd and curd (60.0%, *n* = 9), vegetable oil (31.0%, *n* = 9) and curd drink and yoghurt (26.8%, *n* = 11). Health claims were carried most frequently on labels for dried curd and curd (53.8%, *n* = 7), buckwheat, rice and millet (52.5%, *n* = 21) and curd drink and yoghurt (51.2%, *n* = 21). Higher rates of health claims were found in barley, flax and wheat flour, buckwheat, rice and millet and breakfast cereal.

### 3.3. Types of Health Claims

For most of the nutrient function (*n* = 129 of 176 claims) and other function claims (*n* = 116 of 148 claims), health benefits were related to a whole product or its ingredients, such as “Rye contained in the product supports the digestive system” (nutrient function claim) or “Pure chocolate contained in the product improves brain function” (other function claim) (Table A2 and Table A3). Therapeutic claims were the second most common claims with 160 claims found across the sample. Again, these claims were mostly based on a whole product or its ingredients (Table A4). Reduction of disease risk claims were the least prevalent health claims, identified 39 times across the sample (Table A5).

### 3.4. Credibility of Nutrition and Health Claims

The credibility of the claims was very low. For nutrition claims, this was mostly due to the lack of information about the claimed nutrients in the nutrient declaration or the absence of any nutrient declaration. Overall, 131 claims out of a total 288 nutrition claims (45.5%) had no information on the content of a claimed nutrient, no nutrient declaration or was a general claim. General claims were the claims regarding the high content of vitamins or minerals of a product, without referring to a specific vitamin or mineral. Example of a general claim is “The product is a source of vitamins and minerals”. Only 97 nutrition claims (33.7%) were accompanied by complete and accurate information on the claimed nutrients and their content and thus deemed as credible. For the remaining 60 nutrition claims (20.8%), nutrient content did not meet the established criteria for nutrition content claims from Codex, e.g., the criteria for a “good source of protein” claim is that the product’s protein content should not be less than 10% of the nutrient reference value (NRV) for protein (Table 4).

Even fewer health claims were credible. One-third of all health claims (*n* = 160 of 523 claims) were therapeutic claims, prohibited in the EU. Of the remaining types of health claims (*n* = 363), only 18 claims were found on the list of authorised claims of the EU. Of these, only six claims met the specific criteria of the claims for the nutrient content (Table 5). Claims regulations in the EU authorise claims for specific nutrients/substances or food/food categories, not for the food products carrying the claim [19]. Most of the non-therapeutic health claims on Mongolian products (*n* = 263 of 309 claims) would be disqualified for use in the EU countries as they were based on a whole food product or its ingredients.

Claims that were in the Mongolian language were less credible than claims in other languages. Only 25.4% (*n* = 50/197 claims) of nutrition claims in Mongolian were credible versus 51.6% (*n* = 47 of 91 claims) of the claims in other languages. Nutrient information was not provided for over half of the nutrition claims (53.8%, *n* = 106) in Mongolian compared to 27.5% (*n* = 25) of the claims in other languages (Table 4). There were no health claims in the Mongolian language that met the relevant criteria in the comparison country (Table 5).

### 3.5. Healthiness of Products with Claims

Based on nutrient profiling, 54.2% (*n* = 140) of products with nutrition claims and 40.5% (*n* = 184) of products with health claims were less healthy products (Table 6).

## 4. Discussion

In this study, approximately 10% (*n* = 175) of all products carried health claims and 9% (*n* = 149) carried nutrition claims. The rate of health claims was similar to the findings of other studies from Australia (11%) and South Africa (10.2%) but lower than the prevalence of claims identified in Ireland (17.8%) [17,22,23]. The rate of health claims was higher in Mongolia than previously reported on products from the EU, the USA, Malaysia and Indonesia (0–7.1%) [24,25]. The rate of nutrition claims was much lower than the other countries’ rates [16,23,24,25].

The proportion of unhealthy products with nutrition claims in our study (54.2%) was higher compared to the other studies from Australia, Canada and some EU countries where 29–42% of products carrying nutrition claims had less healthy nutrient profiles [26,27,28]. Likewise, products with health claims were less healthy in our study (40.5% were less healthy) compared to products with health claims in the studies from Australia (31%) and EU countries (30%) [27,28]. In order to prevent unhealthy products to have claims, some countries implement regulations to restrict making claims on certain types of foods or to endorse claims on foods meeting certain nutrient eligibility criteria [8].

This study identified that nutrition and health claims found on food and beverage products in Mongolia had very low levels of credibility. In particular, claims made on products labelled in Mongolian were less credible than claims in other languages. Most health claims were found on Mongolian language products and nearly all of them were not credible. Almost all of the 160 therapeutic health claims were on Mongolian language products. These types of claims are prohibited by Codex Alimentarius and in other countries. This contrasts to other studies, which have reported few cases of such claims on products [17,22,24]. A similar pattern was identified for nutrition claims, whereby only one-third of these claims (33.7%, *n* = 97) were deemed credible. Lack of supporting information on the content of the claimed nutrients (45.5%, *n* = 131 of 288 claims) largely contributed to the low credibility of nutrition claims. This finding is exceptional when compared to other studies. For example, a similar survey from Australia found only 7.2% (*n* = 322) of nutrition claims were not credible [16]. Again, nutrition claims on Mongolian language products were half as likely to be credible than claims on products labelled in other languages.

Such variations in the credibility of claims reflect the status of food labelling regulation in Mongolia and in other countries at the time of the study. A high prevalence of therapeutic claims was also reported in a Serbian study, in which 17% of products had therapeutic claims [29]. At the time of these studies, in both Mongolia and Serbia there was no government regulation on the use of nutrition and health claims, allowing these to be freely used without any independent validation or safeguards. Soon after this survey was conducted, a new Mongolian food labelling standard, MNS 6648:2016, came into force in January 2018 [14]. However, the new standard lacks a clear definition on nutrition and health claims, specification on different types of claims and criteria for making those claims or a substantiation framework for claims, such as minimum criteria for the healthfulness of products bearing a claim. The standard states that claims be approved by an authorised government organisation prior to use, however, a procedure for that has not been developed.

The potential negative impact of claims on food choices and consumption [15,30] can be particularly significant in Mongolia. The results of this study highlight the pervasiveness of poorly regulated food claim practices. In addition, the population has relatively poor levels of nutrition literacy [31] and low awareness on food labelling. The added burden of non-credible claims on less healthy food products may worsen the process of nutrition transition currently underway in Mongolia. Such labelling essentially disseminates misinformation and hinders healthy choices.

The study has several limitations. First, the survey sample does not represent all packaged food products available at the marketplaces in Mongolia. However, using a prior developed list of product categories and an attempt to ensure the representation of domestic and imported products and different brands, the sample captured all common types of packaged products in the marketplace. Second, due to the convenience sampling, calculation of percentages and statistical tests was not possible in some cases due to a small number of claims per comparison group.

## 5. Conclusions

Mongolia is experiencing rapid nutrition transition, similar to many developing nations. Nutrition labelling policy is increasingly important as marketplaces and population diets are being dominated by processed packaged foods. Major issues in the use of nutrition and health claims in Mongolia were identified, whereby most claims were not credible and not based on scientific evidence and many were found on unhealthy products due to the unregulated and voluntary use of nutrition and health claims by food producers. New food labelling regulation has been introduced in Mongolia since data were collected, however specifications on the use of nutrition and health claims remain weak. Given Mongolian consumers’ relative poor nutrition literacy, it is likely that they are at greater risk of the negative effects of misleading claims on their food choices and consumption. Regulations for food claims are in their early stages of development in Mongolia and more rigorous regulations providing clear guidance about the types of permitted claims and conditions under which claims can be made are needed. The current regulations regarding nutrition and health claims are needed to be upgraded in consultation with the Codex guidelines for use of nutrition and health claims as well as claims regulations of other countries. Awareness of consumers and food producers on nutrition and health claims is needed to be improved.

## Figures and Tables

**Figure 1 ijerph-17-07456-f001:**
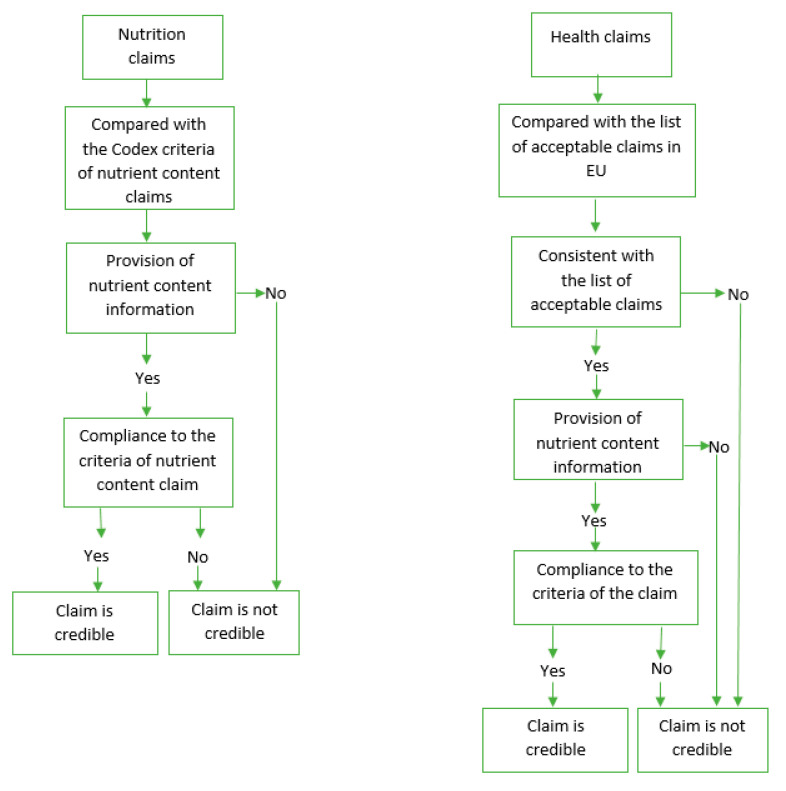
Assessment of credibility of nutrition and health claims.

**Table 1 ijerph-17-07456-t001:** Types of claims.

Type of Claims	Definition		Example of Claim
Nutrient content claim	Claims that describe the level of a nutrient contained in a food		“Source of calcium”; “High in fibre”; “Low in fat”
Health claim	Statement about a relationship between a food or a constituent of that food and health		Examples of health claims are given below.
Type of health claim	Nutrient function claim	Claims that describe the physiological role of a nutrient in growth, development and in maintaining and supporting normal functions of the body (not related to a specific disease)	“Calcium for healthy bones and teeth. Food X is a source of calcium.”
	Other function claim	Claims related to positive contribution of a food or a constituent of that food to health or improvement of a body function. In this study, claims related to substances other than nutrients were classified in this category.	“Fibre contained in the product improves peristalsis. Food X is high in fibre.” “Lignans support colon function. The product contains X grams of lignans.”
	Reduction of disease risk claim	Claims related to the reduced risk of developing a disease or health-related condition.	“Diets high in calcium may reduce the risk of osteoporosis. Food X is high in calcium.”
	Therapeutic claim	Claims related to the beneficial effects of nutrients, substances, ingredients or products for treatment, alleviation or cure of diseases and conditions [8]. These types of claims are prohibited by Codex Alimentarius. Claims relating to the prevention of diseases are considered therapeutic claims as well.	“The product helps in liver diseases.” “Regular consumption of the product prevents cardiovascular diseases.”

**Table 2 ijerph-17-07456-t002:** Food categories covered in the survey.

Food Category	Food Category Code	Products Labelled in		Total	
		Mongolian	Other Languages	*n*	%
Cakes, sweet biscuits and pastries, sweet bakery products	2	188	114	302	17.5
Beverages	4	118	93	211	12.2
(a) Juices	4a	(4)	(28)	(32)	(1.9)
(b) Milk drinks	4b	(30)	(6)	(36)	(2.1)
(c) Energy drinks, tea and coffee	4c	(77)	(54)	(131)	(7.6)
(d) Other sugar-sweetened beverages (juice drinks, soft drinks, flavoured water, etc.)	4d	(7)	(5)	(12)	(0.7)
Chocolate and sugar confectionary, energy bars and desserts	1	28	176	204	11.8
Processed meat, poultry, fish and similar	14	143	40	183	10.6
Processed fruit and vegetables	16	71	98	169	9.8
Fresh or dried noodles, pasta, rice and grains	12	77	40	117	6.8
Sauces, dips and dressings	18	10	91	101	5.9
Savoury snacks (chips, crisps, processed seaweed, crackers, nuts, etc.)	3	23	54	77	4.5
Yoghurt, sour milk, cream, curds	7	55	5	60	3.5
Butter, vegetable oils, other fats	10	8	47	55	3.2
Ice cream	5	26	25	51	3.0
Ready-made and convenience foods and composite dishes	9	8	42	50	2.9
Bread, bread products	11	47	2	49	2.8
Fresh and frozen meat, poultry, fish and similar	13	30	3	33	1.9
Breakfast cereals	6	6	15	21	1.2
Other products ^1^	NA	10	9	19	1.1
Cheese	8	3	13	16	0.9
Tofu products	17	5	0	5	0.3
Total		856	867	1723	100.0

^1^ Other products included products (bottled water, herbal tea, baking powder, infant formula and alcoholic beverages) that are not included in the food categories of the WHO nutrient profile model for the WPR; NA—not applicable.

**Table 3 ijerph-17-07456-t003:** Prevalence of nutrition and health claims.

Type of Claims		Products with at Least one Claim		Total Number of Claims	Median Claims per Product	Rate per 100 Products ^1^
		*n*	% ^1^			
Nutrition claim		149	8.6	288	2.0	16.7
Health claim	Nutrient function claim	114	6.6	176	1.0	10.2
	Other function claim	93	5.4	148	1.0	8.6
	Reduction of disease risk claim	26	1.5	39	1.0	2.3
	Therapeutic claim	79	4.6	160	2.0	9.3
	Total	175	10.2	523	2.0	30.4

^1^ Percentages and rates were estimated for the total number of products of 1723.

**Table 4 ijerph-17-07456-t004:** Credibility of nutrition claims by label language.

Type of Claim	Label Language	Total Number of Claims	Information on the Nutrient				Quantity Statement	
			Not Provided		Provided		Accurate	Inaccurate
			*n*	%	*n*	%	*n*	*n*
Nutrition claim	Mongolian	197	106	53.8	91	46.2	50	41
	Other	91	25	27.5	66	72.5	47	19
	Total	288	131 ^1^	45.5	157	54.5	97	60

^1^ Nutrient information was missing due to lack of nutrient declaration (7.6% of the claims) or no values for the nutrient (for 33.3% of the claims) or was a general claim (for 4.5% of the claims).

**Table 5 ijerph-17-07456-t005:** Comparison of health claims with the authorised claims in the EU.

Type of Claims		Total Number of Claims	Permitted Claims		Credible Clams
			*n*	%	*n*
Nutrient function claim		176	17	9.7%	6
Other function claim		148	1	0.7%	0
Reduction of disease risk claim		39	0	0	0
Therapeutic claim		160	0	0	0
Label language	Mongolian	453	11	2.4%	0
	Other ^1^	70	7	10%	6
Total		523	18	3.4%	6

^1^ “Other” included Russian, English and Korean.

**Table 6 ijerph-17-07456-t006:** Application of nutrient profiling model to the products with nutrition and health claims.

Type of Claims		Label Language	Total Number of Claims	Claims Covered in Nutrient Profiling ^1^		Ranked as			
						Healthy		Unhealthy	
				*n*	%	*n*	%	*n*	%
Nutrition claim		Mongolian	197	175	88.8	79	45.1	96	54.9
		Other	91	83	91.2	39	47.0	44	53.0
		Total	288	258	89.6	118	45.7	140	54.2
Health claim	Nutrient function claim	Mongolian	138	121	87.7	73	60.3	48	39.7
		Other	38	36	94.7	23	63.9	13	36.1
		Sub total	176	157	89.2	96	61.1	61	38.9
	Other function claim	Mongolian	134	117	87.3	67	57.3	50	42.7
		Other	14	14	100.0	7	50.0	7	50.0
		Sub total	148	131	88.5	74	56.5	57	43.5
	Reduction of disease risk claim	Mongolian	37	30	81.1	12	40.0	18	60.0
		Other	2	2	100.0	2	100.0	0	0
		Sub total	39	32	82.1	14	43.8	18	56.2
	Therapeutic claim	Mongolian	144	118	81.9	75	63.6	43	36.4
		Other	16	16	100.0	11	68.8	5	31.2
		Sub total	160	134	83.8	86	64.2	48	35.8
	Total	Mongolian	453	386	85.2	227	58.8	159	41.2
		Other	70	68	97.1	43	63.2	25	36.8
		Total	523	454	86.8	270	59.5	184	40.5

^1^ 30 nutrition claims and 69 health claims could not be assessed against the nutrient profiling model due to lack of a nutrient declaration or missing nutrient information on the declaration.

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
