# Peer review of "Prevalence and Credibility of Nutrition and Health Claims: Policy Implications from a Case Study of Mongolian Food Labels"

_ijerph, 2020, doi:10.3390/ijerph17207456_

Round 1

Reviewer 1 Report

Prevalence and credibility of nutrition and health 3 claims: Policy implications from a case study of 4 Mongolian food labels

The study aimed to determine the use of nutrition and health claims on packaged foods sold in Mongolia and examine the credibility of these claims. The results obtained are very interesting, because food we eat can affect our health in different ways.

All work has been well planned and developed. I found only minor corrections and the text is suitable for publication

Methodology -  list the product categories that were included in the study and their quantity, or write that all products from the store were taken into account

Results Point 3.2 should be included in the methodology

References number 15 and 16 are too old

I do not see in the text of links to the tables in Appendix

Please include the regulation (number, year) that are in force in Mongolia for example „ In the context of EU Regulation 1924/2006, health claims are claims that state, suggest or imply a relationship between a food or food category and health. Examples hereof are function claims, reduction of risk of disease claims, or claims referring to the growth and development of children. Function claims are based on generally accepted scientific knowledge and are called ‘‘Article 13.1 claims”, e.g. ‘‘this product contains calcium; calcium is relevant for the development of strong bone and teeth”.

Author Response

Point 1: Methodology -  list the product categories that were included in the study and their quantity, or write that all products from the store were taken into account

Response 1: A total of 1723 products belonging to 11 major groups and 56 product categories were covered in the study. The major product groups and categories and number of products sampled from each category were shown in the Appendix Table A1. [pg. 11, line 325]. The previous table was replaced with the current table.

Point 2: Results Point 3.2 should be included in the methodology

Response 2: Thank you for this point. We moved the section 3.2 to the methodology section. [pg. 4, line 150]

Point 3: References number 15 and 16 are too old

Response 3: The references number 15 and 16 were the studies from Australia on health and nutrition claims. These studies were the major studies that assessed the prevalence and credibility of the nutrition and health claims made on food products in Australia. As the aim of our study was corresponded with their aims, we compared our results with the findings of these studies and discussed our results largely basing on their findings. Therefore, we would like to maintain these references in the paper.

Point 4: I do not see in the text of links to the tables in Appendix

Response 4: The links to the tables in Appendix can be find in the text in the following positions.

Table A1 – [pg. 3, line 100]

Table A2,3– [pg. 7, line 194]

Table A4 – [pg. 7, line 196]

Table A5 – [pg. 7, line 197]

Point 5: Please include the regulation (number, year) that are in force in Mongolia for example „ In the context of EU Regulation 1924/2006, health claims are claims that state, suggest or imply a relationship between a food or food category and health. Examples hereof are function claims, reduction of risk of disease claims, or claims referring to the growth and development of children. Function claims are based on generally accepted scientific knowledge and are called ‘‘Article 13.1 claims”, e.g. ‘‘this product contains calcium; calcium is relevant for the development of strong bone and teeth”.

Response 5: The food labelling standard that in force in Mongolia were mentioned in the introduction and discussion sections. We added the number and year of adoption to the standard [pg. 2, line 66 and pg. 9, line 269-270]. 

Reviewer 2 Report

The paper addresses an interesting topic and examines the use and credibility of nutrition and health claims in Mongolia. The authors use a database of 1723 food products sold in Ulaanbaatar, Mongolia. Although, there are several important findings, there are some points through the manuscript that needs further clarification. One of my major concerns is related to the objective of the research that it is not stated clearly. Please see my comments below. I hope it contributes to guide you through your revision.   

ABSTRACT

  • 1, lines 15-16: What do the authors imply by this sentence? Are the nutrition and health claims official?
  • 1, lines 20-21: What do you mean by this sentence? What jurisdictions?
  • Before explaining the results the abstract the authors should inform on the methodology used to collect the data and the methods of analyzing them.

INTRODUCTION

  • 1-2, lines 42-45: Could you please provide a more clear definition of nutrition and health claims and some references on these statements?
  • Perhaps I am missing something, but your aim is to measure the use and credibility of nutrition and health claims in 2017 (when you collect your data on nutrition and health claims from processed food [pg. 2, line 80]) but the legislation came to enforcement in 2018 (pg. 2, line 61). Isn’t it expected to find low credibility for nutritional claims and health claims that are lunched in the market perhaps voluntarily without anyone controlling them? Please explain.
  • Could you please provide some more information on this new food labelling standard / regulation on nutrition and health claims?

MATERIAL AND METHODS

  • The data collection part is confusing. The authors need to make changes and better specify this part. For example, did these students go to different supermarkets and/or other type of stores to collect the product information? Please specify.
  • 2, lines 87-90: In this sentence, do you mean that the food categories and sub-categories included in the database (study) were chosen based on the common food purchases of Mongolian households? If yes, then please correct accordingly.
  • 2, lines 90-91: What did you pre-tested in one supermarket?
  • The “coding of photographs” section I think is unnecessary, you may delete it.
  • In the section of “Data Analysis” I got a bit lost. More specifically:
    • 3, line 103-104: what do you mean by “….the rate of claims per 100 proportions of food products….”
    • The term credibility is normally used to measure how trustful is a nutrition and/or health claim (in this case) to somebody (consumers I guessed), in the sense that the claims are based on scientific evidence, that the information is reliable, and it helps consumers distinguish the healthiness of a food product. Either specify what credibility means in your paper or use another term that is easy understandable by a majority of readers. Reliability might be an option or “authorized” versus “voluntary” nutrition and health claims in the Mongolian food market might be another option. Change the title of the paper accordingly.  
    • Then, it is unclear to me whether you had a legislation in Mongolia that controlled the nutrition claims or you used the Codex CAC/GL 23-1997 of FAO. This and also the reason of comparing the health claims found in the Mongolian market with the legislations on the health claims of Australia, EU, Canada and the USA (which change between them) should be specified and be very clear in the introduction. Why not comparing the reliability of your health claims with the just one legislation (e.g. EU) instead of four? Is it because there is a high prevalence of food products from these countries in the Mongolian market? Please clarify. 
  • 3, lines 115-118: Can you please explain the WHO Nutrient profile model in further details on what it does, what is the aim of this model, how can we interpret the codification (so readers can also understand the results from Table 1 and the rest)?

RESULTS

  • 4, lines 133-134: Based on which legislation are these health claims in Box 1 described?
  • 5, lines 138-139: Where there products with two health claims in the same front of pack? That is a bit weird and not common. Perhaps it is better to separate the prevalence of food packages with 1 nutrition claim and 1 health claim and those with more than 1.
  • 5, lines 146-147: Instead of “use” I think the authors mean “prevalence of claims”. Also correct the numbers between claims labelled in Mongolian (n=856) versus other languages (n=867) as they do not correspond with the statement of the sentence.
  • 6, line 159-160: What do you mean by (n=129/176 claims) and (n=116/148 claims)?
  • 6, lines 169-171: Can you be more specific on what where the 131 out of 288 nutrition claims? You state that these products “…had no information on the claimed nutrient, no nutrient declaration or was a general claim without reference to a specific nutrient” then, why do you consider them as products with nutrition claims?
  • 6, lines 176-184: In my opinion, you should focus in comparing the credibility of nutrition and health claims considering only one legislation (e.g. EU) instead of all of them. I would choose the one that represents the highest prevalence in the food market. For example, if the majority of food products is imported from Europe then I would use the EU legislation and vice versa for the rest.

DISCUSSIONS

  • 8, lines 223-227: This should be mentioned and be clarified in the introduction first.
  • 8, lines 234-239: These should be moved to the introduction and tailored there as a justification of this research.
  • Why don’t you mention anything about healthy vs. un-healthy food and nutrition and health claims in this section? It is very interesting to discuss these result and compare it with other studies since the authors also mention it in the results.

CONCLUSIONS 

In the conclusion I would clarify that most on the nutrition and health claims on food products in the Mongolian food market are labelled voluntary by the producers in order to differentiate the healthiness of their products. Yet, the majority of these claims are not labeled based on any scientific evidence and are not controlled by any public authority. This can mislead consumers and lead them in uninformed food choices.

  • What are the policy implications that this research proposes to the government? Educating consumers for example? Homogenizing legislations with the rest? Considering foreign legislations in developing the Mongolian legislation? Etc.

Author Response

Response to Reviewer 2 Comments

ABSTRACT

Point 1: 1, lines 15-16: What do the authors imply by this sentence? Are the nutrition and health claims official?

Response 1: By this sentence we implied that the claims identified on product packs were assessed for compliance with the Codex Alimentarius guidelines and the claims regulations of the EU. At the time of the survey was conducted, there were no regulations regarding nutrition and health claims in Mongolia.

We have changed the sentence as Nutrition claims were examined for their compliance with the Codex Alimentarius guidelines and health claims were assessed against the claims regulations of the EU” [pg 1, line 15-17]

Point 2: 1, lines 20-21: What do you mean by this sentence? What jurisdictions?

Response 2: By this sentence we meant that of identified health claims only a few claims would be permitted in the compared countries by complying the requirements of the regulations for health claims governing in these countries. Jurisdictions were referred to the jurisdictions of claims regulations in the compared countries.  

In order to convey this meaning clearly, we have changed the sentence as “Only a few claims would be permitted in the EU countries by complying with the EU regulations” [pg 1, line 21-22]

Point 3: Before explaining the results the abstract the authors should inform on the methodology used to collect the data and the methods of analyzing them.

Response 3:  The point was addressed and included in the abstract

“A cross-sectional study examined the label information of 1723 products sold in marketplaces in Ulaanbaatar, Mongolia. The claim data was analysed descriptively.” [pg 1, line 14-15.].

INTRODUCTION

Point 4: 1-2, lines 42-45: Could you please provide a more clear definition of nutrition and health claims and some references on these statements?

Response 4: The sentence was changed as follows to provide a clearer definition of nutrition and health claims and a reference was added.

“Nutrition claims state, suggest or imply that a food has particular nutritional properties including but not limited to the energy value and to the content of protein, fat and carbohydrates, as well as the content of vitamins and minerals. Health claims refer to relationships between a food or a constituent of that food and health [6]. [pg 2, line 44-48]

Point 5: Perhaps I am missing something, but your aim is to measure the use and credibility of nutrition and health claims in 2017 (when you collect your data on nutrition and health claims from processed food [pg. 2, line 80]) but the legislation came to enforcement in 2018 (pg. 2, line 61). Isn’t it expected to find low credibility for nutritional claims and health claims that are lunched in the market perhaps voluntarily without anyone controlling them? Please explain.

Response 5: Thank you for raising up this point. We had an idea that nutrition and health claims on products would be of low credibility reflecting the lack of regulations at the time of the study. The purpose of the study was to provide a baseline evaluation for claim labelling practices before the legislation, which will allow evaluations of the outcomes of the regulation in the future. We have highlighted this point now in the paper [pg. 2, line 85-88]. “As food labelling regulations are currently in transition in this country, this study provides a critical baseline evaluation of the food labelling landscape to guide identification of areas of concern and provide a basis for assessing progress on policy implementation.”

Point 6: Could you please provide some more information on this new food labelling standard / regulation on nutrition and health claims?

Response 6: The following sentence was added regarding the regulation on nutrition and health claims that included in the new food labelling standard.

A definition of nutrition claim was provided in the standard, together with the requirement to declare the amount of the claimed nutrient. The standard also introduced the concept of scientific substantiation of health claims. However, the standard does not specify the types of nutrition and health claims that are permitted and lacks requirements regarding criteria for making claims”  [pg 2, line 76-80]

MATERIAL AND METHODS

Point 7: The data collection part is confusing. The authors need to make changes and better specify this part. For example, did these students go to different supermarkets and/or other type of stores to collect the product information? Please specify.

Response 7: We clarified the data collection method by specifying the students visited their usual (not pre-identified) supermarkets and/or grocery stores for data collection and the following sentence was added in this section.

“The students collected the label information of food products from supermarkets and grocery stores located throughout the city. They were instructed to collect the product information from any supermarket or grocery store at their convenience.” [pg 3, line 96-98]

Other changes in this section related to data collection were made in terms of wording and the order of sentences as follows.

  • The first sentence was changed to “A survey of packaged food product labels was conducted in Ulaanbaatar, the capital city of Mongolia, during November and December 2017.” [pg 3, line 93-94]
  • The third and fourth sentences reporting regarding taking photographs of products were moved towards to the end of the section. [pg 3, line 108-111]

Point 8: 2, lines 87-90: In this sentence, do you mean that the food categories and sub-categories included in the database (study) were chosen based on the common food purchases of Mongolian households? If yes, then please correct accordingly.

Response 8: Yes, we meant so as the food categories that we adapted from the household socio-economic survey of the National Statistics Office of Mongolia were based on the common types of food products that consumed by Mongolian households and we also added some common types of packaged food products. 

We corrected the sentence as follows.

These food categories and subcategories were based on the food categories’ classification used in the household socio-economic survey of the National Statistics Office of Mongolia [17], which represented the common types of food products used by Mongolian households with some modifications to include other common types of processed food products.”  [pg 3, line 100-103]

Point 9: 2, lines 90-91: What did you pre-tested in one supermarket?

Response 9:  We pre-tested the pre-defined food categories in one supermarket to ensure they were comprehensive. In order to make this point clear, we added some clarification in the existing sentence as follows.

The pre-defined food categories were pre-tested in one supermarket by crosschecking them against the products placed on the shelves in the supermarket and missing food categories were added.” [pg 3, line 104-106]

Point 10: The “coding of photographs” section I think is unnecessary, you may delete it.

Response 10:  We deleted the subheading for the section and shortened the paragraph, and combined it with the previous “Data collection” section and renamed it as “Data collection and coding”. [pg 3, line 112-115] We felt that some information on coding of the photographs was necessary to allow for study replication.

In the section of “Data Analysis” I got a bit lost. More specifically:

Point 11: 3, line 103-104: what do you mean by “….the rate of claims per 100 proportions of food products….”

Response 11:  By “the rate of claims per 100 products” we meant a number of claims identified per 100 products. By this parameter, we wanted to measure the density of occurrence of claims in products. To explain the parameter, we added a phrase “a number of claims per 100 products” in the corresponding sentence.  [pg 3, line 119]  

Point 12: The term credibility is normally used to measure how trustful is a nutrition and/or health claim (in this case) to somebody (consumers I guessed), in the sense that the claims are based on scientific evidence, that the information is reliable, and it helps consumers distinguish the healthiness of a food product. Either specify what credibility means in your paper or use another term that is easy understandable by a majority of readers. Reliability might be an option or “authorized” versus “voluntary” nutrition and health claims in the Mongolian food market might be another option. Change the title of the paper accordingly.  

Response 12:  We would like to maintain the term credibility as it connects with trustworthiness and reliability of claims to consumers in terms of providing reliable information which is based on scientific evidence.

We have clarified the meaning of credibility for this paper and provided an explanation for the term as follows.

By the credibility of claims we perceived trustworthiness and reliability of claims in terms of providing reliable and scientific based information to consumers, as well as providing supporting information on the content of claimed nutrients to back up the claimed nutritional characteristics or health effects of a product.  [pg 3, line 121-124]

Point 13: Then, it is unclear to me whether you had a legislation in Mongolia that controlled the nutrition claims or you used the Codex CAC/GL 23-1997 of FAO. This and also the reason of comparing the health claims found in the Mongolian market with the legislations on the health claims of Australia, EU, Canada and the USA (which change between them) should be specified and be very clear in the introduction. Why not comparing the reliability of your health claims with the just one legislation (e.g. EU) instead of four? Is it because there is a high prevalence of food products from these countries in the Mongolian market? Please clarify. 

Response 13:  At the time of the survey, in 2017, there were effectively no regulations in Mongolia that controlled nutrition and health claims. There was a standard on nutrition labelling MNS CAC GL 2:2007 that was in effect at the time of the survey in 2017. It was a 1:1 translation of the Codex guidelines on nutrition labelling CAC GL 2-1985 and was implemented on voluntary basis. However, the standard was not able to provide regulations for nutrition and health claims due to its voluntary nature and lack of specific guidance on nutrition and health claims. The guideline had a few clauses providing a definition of nutrition claim and on declaration of the content of a claimed nutrient in the case of nutrition and health claims, copying to its original Codex document. However, the translations of the document were largely incorrect, with the requirements for claims mixed up with the requirements for nutrient declarations and thus it was not able to provide proper guidance on claims. Even though the problem related to poor translation was fixed in the new Mongolian food labelling standard of 2018, and a clearer definition of nutrition claims was provided and authorisation of health claims was addressed in the new standard, the new standard still lacked a definition of health claims, types of nutrition and health claims, and criteria for making nutrition and health claims. This deficit of the new standard was mentioned in the discussion section of the paper [pg 9, line 272-276]. These were the main reasons of using the Codex CAC/GL 23-1997 and legislations on health claims of the EU instead of using the previous or current Mongolian food labelling standard in our analysis of credibility of claims.

We considered this point and added some clarification on the previous guideline on nutrition labelling in Mongolia in the introduction section as follows.

Prior to this there was effectively no regulation relating to nutrition and health claims on food packages. The previous guideline on nutrition labelling of 2007, which was an apparent translation of the Codex guidelines on nutrition labelling [5], lacked capability to provide proper regulation due to its poor translation (introducing errors) and voluntary nature”.  [pg 2, line 67-71]

Also we added clarification in the data analysis section on why we used the Codex guidelines and the claim regulation of EU in our analysis.

“The Codex guidelines and the claims regulation of EU were used in the credibility analysis of claims as the current national food labelling standard (2018) did not contain criteria for making nutrition and health claims.” [pg 3, line 124-126].

We tried to look at a range of policies that are available globally and assess the identified health claims against these policies. Based on the reviewer comments, we decided to compare with only the EU regulation considering relatively high prevalence of food products from the EU in the Mongolian market compared to other countries compared and similar results of the comparison between the compared policies. We have amended the corresponding parts in the data analysis section by omitting the comparisons with other countries than the EU and adding an explanation why we used the EU regulation [pg 3, line 130-133], as well as in the results [pg 7-8, line 211-228] sections by omitting the comparisons with countries other than the EU.

Point 14: 3, lines 115-118: Can you please explain the WHO Nutrient profile model in further details on what it does, what is the aim of this model, how can we interpret the codification (so readers can also understand the results from Table 1 and the rest)?

Response 14: The model was explained in more details including its purpose and use, as well as the food categories that are included in the model [pg 4, line 139-146].

RESULTS

Point 16: 4, lines 133-134: Based on which legislation are these health claims in Box 1 described?

Response 16: The definitions of health claims were based on the Codex standard CAC/GL 23-1997, except therapeutic claims. We added a reference for the definition of therapeutic claims. We moved the subsection of classification of claims to the methodology section [pg 5, line 152].   

Point 17: 5, lines 138-139: Where there products with two health claims in the same front of pack? That is a bit weird and not common. Perhaps it is better to separate the prevalence of food packages with 1 nutrition claim and 1 health claim and those with more than 1.

Response 17:   It was common for the products to have more than one health claim per product at the same time. Example of a product with multiple health claims: “X product is suitable to consume during digestive discomfort, as well as it prevents the development of cardiovascular diseases and hypertension.” In order to demonstrate this point, we calculated median claims per product (Table 2) and also provided the proportions of products with more than one nutrition and health claim [pg 6, line 169-173].

Point 18: 5, lines 146-147: Instead of “use” I think the authors mean “prevalence of claims”. Also correct the numbers between claims labelled in Mongolian (n=856) versus other languages (n=867) as they do not correspond with the statement of the sentence.

Response 18: …The “use” was changed to “prevalence of claims”. Also in the text, previously stated phrase “…between two and 22 times…” was changed to “…2.2 and 21.7 times…”   [pg 6, line 178].

Point 19: 6, line 159-160: What do you mean by (n=129/176 claims) and (n=116/148 claims)?

Response 19: By n=129/176 claims we meant 129 claims out of 176 claims. We changed it as n=129 of 176 claims. [pg 7, line 191-220].

Point 20: 6, lines 169-171: Can you be more specific on what where the 131 out of 288 nutrition claims? You state that these products “…had no information on the claimed nutrient, no nutrient declaration or was a general claim without reference to a specific nutrient” then, why do you consider them as products with nutrition claims?

Response 20:   Lack of supporting information on the claimed nutrients was common for nutrition claims and largely contributed to the low credibility of nutrition claims. We found that 45.5% (n=131 of 288 claims) of the nutrition claims had no supporting information on the content claimed nutrients or was a general claim. To make the point clearer, we now highlight this in the discussion [pg 9, line 258-260], as well as included a description on what we meant by a general claim in the results section and provided an example of such claim. Example of a general claim: “The product is a source of vitamins and minerals.” [pg 7, line 202-205].

Also, we changed “131 out of 288 nutrition claims” to “131claims out of 288 nutrition claims”, as well as “…had no information on the claimed nutrient” to “… had no information on the content of a claimed nutrient” [pg 7, line 201-202].

Point 21: 6, lines 176-184: In my opinion, you should focus in comparing the credibility of nutrition and health claims considering only one legislation (e.g. EU) instead of all of them. I would choose the one that represents the highest prevalence in the food market. For example, if the majority of food products is imported from Europe then I would use the EU legislation and vice versa for the rest.

Response 21: We changed the results of the credibility analysis by leaving only the EU comparison [pg 7, line 211-216, 228 Table 4], as described above.

DISCUSSIONS

Point 22: 8, lines 223-227: This should be mentioned and be clarified in the introduction first.

Response 22:   This point was included and clarified in the introduction [pg 2, line 67-71].

Point 23: 8, lines 234-239: These should be moved to the introduction and tailored there as a justification of this research.

Response 23: The points regarding the transition to the market economy and low familiarity of the population with food labelling were moved to the introduction [pg 2, line 57-60].

Point 24: Why don’t you mention anything about healthy vs. un-healthy food and nutrition and health claims in this section? It is very interesting to discuss these result and compare it with other studies since the authors also mention it in the results.

Response 24: We included a discussion around this point in the discussion section [pg 8,9, line 243-250].

CONCLUSIONS 

Point 25: In the conclusion I would clarify that most on the nutrition and health claims on food products in the Mongolian food market are labelled voluntary by the producers in order to differentiate the healthiness of their products. Yet, the majority of these claims are not labelled based on any scientific evidence and are not controlled by any public authority. This can mislead consumers and lead them in uninformed food choices.

Response 25:   We reflected the point in the conclusions. [pg 10, line 292-294].

Point 26: What are the policy implications that this research proposes to the government? Educating consumers for example? Homogenizing legislations with the rest? Considering foreign legislations in developing the Mongolian legislation? Etc.

Response 26:  We specified the policy implications of the study further and highlighted the need on advancing the current regulations based on the advanced policies of other countries as well as the Codex standard on nutrition and health claims. Also consumer and food producer education on the claims was suggested. [pg 10, line 300-303].

Round 2

Reviewer 2 Report

I would like to congratulate the authors. They made a very good job and high effort in alleviating all my concerns, which now reads well. Results are coherent with a clear outcome and significant findings. Please address the following minor changes:

ABSTRACT

Pg. 1, line 17: Please define the acronym EU when it is used for the first time.

Pg. 1, lines 16-17: here I would write: ….while in the absence of a local regulation, the credibility of the health claims were assessed by the use of the European Union (EU) regulation No. 1924/2006 (if this is the case)

INTRODUCTION

Pg. 2, line 77: please refer to the standard as labelling standard MNS 6648:2016 instead of just standard. It is easier for readers to follow you.

DATA ANALYSIS

Pg. 3, line 122: “…scientific evidence based information….”
